# Obesity and Diet Predict Attitudes towards Health Promotion in Pre-Registered Nurses and Midwives

**DOI:** 10.3390/ijerph182413419

**Published:** 2021-12-20

**Authors:** Holly Blake, Kathryn Watkins, Matthew Middleton, Natalia Stanulewicz

**Affiliations:** 1School of Health Sciences, Faculty of Medicine and Health Sciences, University of Nottingham, Queen’s Medical Centre, Nottingham NG7 2HA, UK; 2NIHR Nottingham Biomedical Research Centre, Nottingham NG7 2UH, UK; 3Nottingham University Hospitals NHS Trust, Queen’s Medical Centre Campus, Nottingham NG7 2UH, UK; kathryn.watkins@nuh.nhs.uk; 4School of Medicine, Faculty of Medicine and Health Sciences, University of Nottingham, Queen’s Medical Centre, Nottingham NG7 2UH, UK; mzymm16@nottingham.ac.uk; 5Faculty of Health and Life Sciences, School of Applied Social Sciences, De Montfort University, Leicester LE1 9BH, UK; natalia.stanulewicz@dmu.ac.uk

**Keywords:** obesity, diet, nurses, midwives, health promotion, public health

## Abstract

Nurses and midwives are integral to public health promotion; in the UK, they are advised to act as role models by their governing body, but overweight or obesity rates are high. We explored whether obesity and dietary habits are related to attitudes towards healthy role modelling and health promotion practice. A total of 346 pre-registered UK nurses and midwives (92.6% female; 18–53 years) completed an online survey. Items included body composition, dietary habits assessed by the Lifestyle Behaviour Questionnaire (LBQ), attitudes towards being role models for health (RA: role attitudes) and attitudes toward health promotion practice (ATHPP): 33.8% of the sample self-reported as overweight or obese; 67.6% did not consume 5-a-day portions of fruit/veg; 31.5% reported a healthy diet; and 89.6% believed their diet could be healthier. Positive RA was significantly linked to health promotion engagement (HP) (ß = 0.33, *p* < 0.001). Positive ATHPP was significantly predicted by lower BMI (ß = −0.08, *p* = 0.078), positive RA (ß = 0.67, *p* < 0.001), lower HP (ß = −0.25, *p* < 0.001) and male gender (ß = 0.09, *p* = 0.02). Greater confidence in patients valuing healthcare professional’s advice was predicted by healthier diet (ß = 0.11, *p* = 0.03), lower BMI (ß = −0. 16, *p* < 0.01), more positive RA (ß = 0.14, *p* < 0.01) as well as HP engagement during training (ß = 0.20, *p* < 0.01). One’s own motivation to promote health, similarly to ATHPP, was predicted by RA (ß = 0.17, *p* = 0.001) and previous HP engagement (ß = 0.39, *p* < 0.001). Findings show that overweight and obesity are prevalent in pre-registered nurses and midwives; the majority did not consume a healthy diet. Individual’s body composition, diet and attitudes towards role modelling are positively associated with their attitudes towards, and confidence in, health promotion practice. Experiences of health promotion practice during training can have either a positive or a negative influence on attitudes. Mentors and educators could actively promote healthy lifestyles for pre-registered nurses and midwives and facilitate more opportunities for health promotion practice during placements, which includes time for reflection.

## 1. Introduction

Global obesity levels have doubled in the last 30 years [1]. Due to its strong links with chronic illnesses [1,2], obesity is the most widespread threat to health and wellbeing in the UK [3]. The economic burden of obesity is significant, demanding 5–6% of the annual health budget, which is expected to equate to £9.7 billion by 2050 [4,5]. In the 2018 Health Survey for England, 67% of men and 60% of women were classified as overweight or obese, this has increased by 2% since 2014 for both groups [2]. It has been estimated that obesity in nurses in the United Kingdom (UK) is higher than that of the general population. Studies have shown that around a quarter of nurses are obese [6], with the prevalence of obesity amongst nurses at higher levels (at 25.1%) than among other healthcare workers (14.4%) and non-health related occupations (23.5%) [7].

Despite this, nurses and midwives are advocated as leaders in initiating health change [8], and they are also seen as integral to the UK Department of Health’s strategy to tackling obesity [3], providing encouragement and support for healthier lifestyle at a local level. The UK government has put all health care professionals including nurses and midwives at the forefront of health promotion policies including obesity prevention, via healthy diet promotion as part of their 2018 prevention strategy [9,10,11]. Aligned with their professional responsibility, nurses and midwives are therefore strongly encouraged to act as role models as per guidance from the UK Nursing and Midwifery Council (NMC) [12].

An integrated definition of ‘role models’ defines them as behavioural models, representing the possible and being inspirational [13]. Role models play an important part not only in promoting health behaviours to patients, but also health care professionals. For student (pre-registered) nurses and midwives, clinical and academic mentors help to form their professional identity and by emulating the professional behaviour of those above them, students transition into professional roles [14]. Although nurses generally perceive themselves to be important role models for health behaviours [15] as do pre-registered nurses [16], some may lack the awareness of this expectation from their professional body. Lack of support during the transition from student to qualified health professionals, poor education on how to become effective role models to the public, and the challenges of making healthy lifestyle choices in both education and placement settings have also been noted [15].

Whether nurses and midwives should model their own advice is a controversial debate. Good communication with patients is important in inducing health change, and should be done respectfully and non-judgmentally [17,18]. Some nurses report feeling more positively toward overweight/obese patients and having a better rapport while delivering weight management advice if they themselves were overweight [19] or obese [20]. However, overweight or obese health professionals also report lower confidence levels and feelings of hypocrisy, while delivering weight management advice [15], which may compromise their ability to promote health effectively. This suggests that role modeling may be important for patient care and strategies for achieving this (e.g., time management, self-care, healthy eating, exercise, managing shifts and sleep) could be made more prominent in healthcare curricula, and in orientation to courses and placements by academic tutors and mentors.

Role-modelling health behaviours is known to be a key strategy in facilitating behavioural change [21]. Multiple studies have shown that weight and appearance can influence nurses’ perceptions of their ability to act as positive role models for health behaviours [16,19,22,23]; a finding mirrored in samples of pre-registered nurses [16]. More importantly, research shows that health professionals who do not heed their own health advice may be less likely to promote healthy lifestyle choices [24,25] and perceive patients to be less likely to follow their guidance [16,26], suggesting that there may be impacts not only for the individual health and wellbeing of nurses and midwives, but also on the provision and receipt of care. This warrants further exploration.

The need for health promotion is rapidly increasing, and in the UK, a national initiative called ‘Make Every Contact Count’ (MECC) advocates that all health and social care staff use the opportunities arising during their routine interactions with patients to have brief conversations on how they might make positive improvements to their health or wellbeing. For the purpose of public health promotion, it is therefore essential that next generation healthcare professionals have positive views towards health promotion, and actively engage in the practice of promoting health to patients and their families. There is a rising prevalence of obesity in nurses and midwives [27,28] and also healthcare students [16,29,30,31]. Weight gain during university years is common [32] and has been linked to poor eating habits, stress and lower physical activity following the transition from a structured secondary education environment to university [33,34]. However, despite higher levels of education and training related to public health and health promotion, healthcare students appear to have similar or even higher rates of overweight and obesity ([16]: 24%; [29]: 44%; [30]: 27.1%; [31]: 28%) compared to university students more broadly. For example, in [35], 22% of students were overweight or obese, (22 countries, *n* = 15,746); in [32], 14.5% of female students were overweight or obese (from 23 universities in England). This is important since weight gain in younger years has been linked to overweight and obesity in adulthood [36]], which poses a significant threat to individual health and wellbeing [1,2,3], and the costs associated with obesity are linked to workplace injury, lost productivity and sickness absence [37,38], which may impact on health and care services once pre-registered individuals join the health workforce.

Surprisingly though, few studies have explored the views of next generation nurses and midwives towards obesity and dietary habits, and whether these relate to their attitudes, perceptions, or experiences of health promotion practice.

Therefore, the aims of this study were: (1) to investigate self-reported/perceived obesity and dietary habits in pre-registered nurses and midwives, and (2) to explore the relationship between these lifestyle factors (overweight and dietary habits) and attitudes towards role modelling and health promotion practice.

## 2. Methods

### 2.1. Ethical Approval

Ethical approval for the study was obtained from the local institutional ethics review board (Ref: SHS-KM-15022016).

### 2.2. Participants

In the UK, student nurses and midwives in training are referred to as ‘pre-registered’; their qualified counterparts referred to as ‘registered’. All pre-registered nurses and midwives studying at a single UK institution were invited to take part in an online survey (*n* = 1335; 1173 nurses and 162 midwives). Eligible participants were required to have experienced over four weeks of health promotion practice in order to have gained sufficient experience in undertaking or observing it.

### 2.3. Procedure

#### 2.3.1. Measures

The survey content was developed by a nurse and a health psychologist. Content was reviewed by five nurse educators, and pilot-tested with five pre-registered nurses and midwives, following which ambiguity in question wording was addressed. The survey included demographic information, height, weight, dietary habits and attitudes towards health promotion (ATHPP). Demographic details collected included age, gender, ethnicity, marital status and year of study. Participants were asked to estimate their weight category to indicate ‘perceived weight’ (underweight; normal weight; overweight; obese). Height and weight were used to calculate body mass index (BMI = kg/m^2^) [39]. For the purpose of this study, participants were classified using BMI to determine ‘reported weight’ in the following ways: underweight: <18.50; normal weight: 18.50–24.99; overweight: 25–29.99; or obese: ≥30, according to World Health Organization (WHO) classification [40].

To assess participants’ views of their own diet and weight, they were asked to rate the following statements on a 4-point Likert scale (strongly agree, agree, disagree, strongly disagree): I am happy with my current weight; I currently eat a healthy diet; My diet could be healthier. With regards dietary habits, 11 items from the Lifestyle Behaviour Questionnaire were utilised [41]. The LBQ was developed from the ‘Behavioural Risk Factor Surveillance System Survey Questionnaire’ [42] and the ‘College Lifestyle and Attitudinal National Survey’ [43]. The items were adapted for use with healthcare students and verified by an expert panel. As in Deasy and colleagues [41], participants’ responses to the ‘dietary habits’ items were summarised to create a diet profile, which was used as a measure for dietary healthiness. A score ≥34 represented a healthy diet, between 23–33 a moderate diet and ≤22 an unhealthy diet.

Participants were also asked three attitudinal questions using items developed by the study authors: (i) Promoting a healthy diet to patients is part of a nurse/midwife’s role; (ii) Nurses/midwives (as appropriate to your profession) should be role models for their patients by maintaining a healthy diet; and (iii) Nurses/midwives (as appropriate to your profession) should be role models for their patients by maintaining a healthy weight) to establish whether participants believed that health promotion was important to the role of nurses or midwives. Cronbach’s alpha for all items was 0.79. Responses were summed to give a Role Attitude (RA) score.

Participants were then asked a series of 13 attitudinal questions (Appendix A) to establish how positively they felt toward promoting a healthy diet. Response options included strongly disagree (1); disagree (2); agree (3); and strongly agree (4). Responses were summarised to give an attitude towards health promotion (ATHPP) score.

Additionally, participants were asked about their experience in observing others engaging in health promotion (HP observation) or having engaged in health promotion themselves (HP engagement). These items were: HP observation: ‘I have witnessed nurses/midwives (as appropriate to your profession) promoting healthy eating on my placements so far)’, HP engagement: ‘I have personally engaged in promoting healthy eating on my placements’. Finally, participants were asked to list the key factors that influenced their dietary habits. Free text response boxes were included to enable participants to expand on their answers. Free-text comments were analysed using thematic analysis [44], employing an inductive approach in which coding and theme development were driven by the content of the comments. This involved analysis of semantic content of the entire free-text data (i.e., whether comments contained references to behaviours, psychological, social or environmental factors) and whether comments were of a positive or negative nature. A coding taxonomy was produced for sorting qualitative data into categories of participant experience, developed by one researcher (KW) in collaboration with a further researcher (HB). Once the main taxonomy had been established (i.e., it accommodated the majority of comments without need for additional categories), the coding of comments was checked by two researchers (KW and HB) and any conflicts were resolved by discussion. Individual categories were then subjected to a second stage of more detailed sorting into subcategories. For example, behavioural factors were categorised into ‘drinking alcohol’ and ‘watching tv’; psychological factors were categorised into ‘stress’, ‘mood’, ‘worry’, and so on. Categories/subcategories were subjected to cross-sectional analysis to highlight the frequencies of common themes and allowed for comparison across groups that reported healthy versus unhealthy diet.

#### 2.3.2. Process

Data were collected between March and May 2016 using a secure online survey hosted on Bristol Online Survey (BOS). All pre-registered nurses and midwives at the participating institution were sent a link to the online survey by email, and the link was also advertised on the student intranet, noticeboards, and relevant social media. Three reminders were posted by the same methods at fortnightly intervals. Interested participants clicked on the survey link and were directed to an online participant information sheet, and the survey questions. Implied consent was assumed from the completion of the survey. Since response rates can be low in online surveys [45], a prize draw incentive of £50 was offered to maximize the response rate. Participants opting to enter the prize draw were asked to provide their email address. These were separated from questionnaire responses prior to analysis, and a prize draw winner was randomly selected on closure of the survey, after which participant contact details were destroyed. Data were analysed using SPSS PASW for Windows Version 25.0 (Armonk, NY: IBM Corp). Analysis included descriptive statistics, Spearman’s and Pearson’s correlations, Chi-Squared Test and linear regression models.

## 3. Results

Of the 1335 pre-registered nurses and midwives who were sent the survey link, 346 opted to complete the online survey (25.9% response rate). Of these, four responses were excluded since these participants were not pre-registered nurses and midwives and so were not eligible for the study. Respondents identified as female (*n* = 319, 92.2%), male (*n* = 26, 7.5%) and non-binary (*n* = 1, 0.3%). Age ranged from 18–53 years (M = 24.27, SD = 6.52), and most of the participants were White British/Irish or of other White background (*n* = 303, 87.7%). Respondents included 310 (89.6%) nurses and 36 (10.4%) midwives, predominantly of single status (*n* = 286, 82.7%). Due to the large difference in group sizes, the results are reported as a single group of pre-registered healthcare professionals.

### 3.1. Weight Status

Of the sample, 33.8% (*n* = 117) (Table 1) reported weight and height that showed that they were either overweight or obese according to BMI classifications. BMI ranged from 14.63 to 51.37 kg/m^2^ (M = 24.50 kg/m^2^, SD = 4.84). There was a significant positive correlation between participants’ perceived weight category (i.e., describing oneself as underweight, healthy weight, overweight or obese; here 109 (31.5%) described themselves as overweight or obese), and their BMI classification based on the reported weight and height (Spearman’s r = 0.80, *p* < 0.001) suggesting that students had a broadly accurate perception of their weight status. However, more than half of participants stated that they were unhappy with their current weight (58.4%, *n* = 202). This weight satisfaction was also highly, and negatively correlated with one’s BMI, in that students with a higher BMI were less satisfied with their weight (r = −0.51, *p* < 0.001).

### 3.2. Dietary Habits

Of the sample, 67.6% (*n* = 234) reported that they did not always consume five portions of fruit and/or vegetables per day (‘5-a-day’). Mean scores for LBQ Dietary Habits ranged from 1–3.67 (M = 2.32, SD = 0.44). Very few participants could be classified by their LBQ score as having a healthy diet (i.e., score > 2.5; healthy: 31.5%, *n* = 109; unhealthy: 68.5%, *n* = 237). These two groups differed significantly in their RA (t(344) = −2.02, *p* < 0.05), weight satisfaction (t(344) = −3.67, *p* < 0.001), and HP attitude (t(344) = −2.34, *p* < 0.05), with those with healthier diet scoring higher on all these outcomes. Of respondents, 72% (*n* = 249) believed their diet to be healthy. That stated, the majority (89.6%, *n* = 313) believed that their diet could be healthier. There was a significant negative correlation between diet profile score and both self-reported BMI (r = −0.16, *p* = 0.003) and perceived BMI (r = −0.12, *p* = 0.026), indicating that those with a less healthy diet profile were more likely to be overweight or obese (including both estimations of their weight category, and calculated BMI from self-reported weight/height).

Participants listed the factors that they felt influenced their dietary habits (Table 2). The most commonly reported factors were: being stressed (78.3%, *n* = 271), one’s mood (77.7%, *n* = 269), boredom (59.8%, *n* = 207) and money issues (56.9%, *n* = 197). Participants who had a less healthy dietary habits reported ‘not having enough money’ (*p* = 0.001), ‘exam pressure’ (*p* < 0.03) and ‘worry’ (*p* < 0.03) influenced their diet significantly more often than other factors (see Table 2). There was also a trend for the number of students mentioning health as one of the factors affecting their diet more often in the healthier diet group (although numbers here are very small). The two groups also differed in agreement regarding the difficulties in having a healthy diet while at university (t(238.93) = 5.56, *p* < 0.001) and during placements (t(344) = 4.77, *p* < 0.001), with the less healthy diet group seeing those as bigger barriers (University: M = 2.40, SD = 0.69; Placements: M = 3.26, SD = 0.78) than those with a healthier diet (University: M = 1.99, SD = 0.60; Placements: M = 2.82, SD = 0.86).

Open-ended responses from participants highlighted that working in a clinical area was perceived to have a particularly negative influence on their dietary habits. Participants reported that they lacked time for healthy eating and home cooking whilst on placement, and the lack of routine and fatigue experienced when on shifts negatively influenced their eating habits.

### 3.3. Role Attitude (RA)

Participants were asked about their experience and/or observation of health promotion practice. Around two-thirds of participants had actively engaged in (68.8%, *n* = 238), or witnessed (63.6%, *n* = 220), health promotion practice on placements. However, the remainder had never observed (36.4%, *n* = 126) or engaged (31.2%, *n* = 108) in health promotion at all. Of those who had never engaged in, or observed health promotion practices, nearly half were in their final year of training (47.5%, *n* = 38).

Almost all participants (98.6%, *n* = 341) agreed that health promotion was part of the nurse and midwife professional role. The majority of participants (84.7%, *n* = 293) also agreed that nurses and midwives should personally maintain a healthy diet, and over three-quarters (76.9%, *n* = 266) felt that nurses and midwives should maintain a healthy weight. Those three outcomes were averaged to create the RA variable (M = 3.20, SD = 0.51; Cronbach’s alpha = 0.70), which was entered as an outcome variable in a linear regression model (see Table 3), with demographics, BMI, engagement in HP, observation of HP, and healthy diet, as predictors. This model explained 9% of variance in RA (F(8,334) = 4.22, *p* < 0.001), with the only significant predictor being one’s own previous engagement in HP (β = 0.33, *p* < 0.001). This shows that those who engaged more in HP had a more positive RA (i.e., students who practiced health promotion, perceived it as part of the role, and perceived maintaining a healthy diet and weight as being important for it).

### 3.4. Attitude towards Health Promotion (ATHPP)

The scores for ATHPP ranged from 1–4 (M = 2.80, SD = 0.52). Linear regression was used to determine which factors predicted ATHPP (see Table 4), with demographics, BMI, engagement in HP, observation of HP, RA and healthy diet, entered as predictors. This model explained 44% of variance in ATHPP (F(9,333) = 29.58, *p* < 0.001). More positive ATHPP was predicted by lower (at a trend level) BMI (β = −0.08, *p* = 0.078), higher RA (β = 0.67, *p* < 0.001), lower engagement in health promotion practice (β = −0.25, *p* < 0.001) and gender (β = 0.09, *p* = 0.02; with males showing more positive ATHPP). Therefore, overweight or obese participants, those who had lower RA, those who engaged more in HP, and of female gender, were more likely to have a more negative attitude towards health promotion.

Almost two thirds of participants felt confident that patients would value their dietary advice (64.4%, *n* = 223). The same regression model was used to significantly predict participants’ view on patients’ valuing their healthy eating advice (R^2^ = 18%, F(9,333) = 8.22, *p* < 0.001). The results (Table 5) showed that higher confidence in this was linked to lower BMI (β = −0.16, *p* = 0.003), higher engagement in HP (β = 0.20, *p* = 0.002), having a healthier diet (β = 0.11, *p* = 0.03), and having a higher RA (β = 0.14, *p* = 0.006).

Lastly, the same model (Table 6) was used to predict one’s HP behaviour (i.e., using opportunities to promote and advise on healthy diet). The model was significant and explained 23% of variance (F(9,333) = 10.96, *p* < 0.001). As with the model predicting HP attitude, HP behaviour was significantly predicted only by one’s own HP engagement (β = 0.39, *p* < 0.001) and RA (β = 0.17, *p* = 0.001). Thus, promoting healthy eating was more likely among those with more previous HP engagement and more positive RA.

## 4. Discussion

The aims of the study were to investigate obesity and dietary habits in pre-registered nurses and midwives, and to explore the relationship between these lifestyle factors, attitudes towards role modelling and health promotion practice in this group. This study showed that overweight and obesity were prevalent in pre-registered nurses and midwives; two thirds did not consume five portions of fruit and/or vegetables per day, most participants reported an unhealthy diet, and most believed that their diet could be improved.

Those who were overweight or obese were less likely to believe that nurses or midwives should be role models for health and were more likely to have negative attitudes towards health promotion practice. Conversely, those who had a healthy diet, a healthy weight and were more actively involved in health promotion held more positive attitudes towards health promotion practice. They were also more confident that patients would heed their advice than those who were overweight or obese, had poor dietary habits or engaged less in health promotion practice.

While the majority believed that health promotion was part of a nurse or midwife’s role, around one third claimed to have had no involvement in health promotion at all (practice or observation), half of whom were in their final year of training. This suggests that health promotion needs to be more prominent in educational curricula, and opportunities to promote health need to be better facilitated on practice placements.

### 4.1. Dietary Habits and Obesity

More than one third of pre-registered nurses and midwives self-reported as overweight or obese (33.8%), which is lower than population figures (UK national average 63% [46]; 67% of men and 60% of women [47]) and lower than rates observed in healthcare workers (health professionals 61.3% [7]; hospital employees 43% [48]; registered nurses and midwives 50–69% [7,49,50]). The lower rates of overweight and obesity in these student groups compared with healthcare professionals may simply reflect the age of the population as predominantly (although not all) young adults. However, obesity rates are clearly higher in registered nurses and midwives, which creates an argument that preventative approaches are needed during the training years, particularly given that a significant proportion of student healthcare professionals are already overweight or obese (and therefore with associated health risks) at that stage.

Less than one-third (31.5%) of our sample reported having a healthy diet. The UK government recommends consuming five portions of fruit and vegetables (‘5-a-day’) and 78–97 g of fat per day [51]. Around two-thirds (67.6%) of pre-registered nurses and midwives reported that they did not consume five portions of fruit or vegetables per day, and a similar proportion reported that they had consumed more fat, sugar, and convenience foods since starting university. Non-adherence to fruit and vegetable intake recommendations has been shown previously in pre-registered nurses and midwives (78.2–74.1% of nurses ‘rarely’ or ‘never’ eat their ‘5-a-day’ [16,30]), and in the general population (71% of men and 68% of women do not eat their ‘5-a-day’ [47]). These data are concerning given that health literacy is required in student healthcare professionals to support the development of a competent workforce [52]. Coupled with the potential for overestimation of healthiness due to the risks of social desirability in self-reporting [53,54], actual figures may be higher than those reported here, stressing the issue even further.

Participants in this sample commonly attributed poor dietary habits to the challenges of working shifts on clinical placements (e.g., impacts on sleep, knowing what to eat and when, access to healthy food). Indeed, shift workers are more likely to be an unhealthy weight [55,56,57,58], with night workers having an increased risk of obesity [59,60]. Many nurses and midwives experience disrupted sleep when working shifts. Short sleep duration, along with other dimensions of poor sleep, has been associated with obesity both cross-sectionally and longitudinally [61]. Shift work and the desynchronisation of circadian rhythms may have adverse effects on nurses’ health—a systematic review [62] showed that night work involves a greater risk of cardiovascular and metabolic diseases compared to daytime work, which may predispose them to obesity. Indeed, shift work has numerous risks for nursing staff, including psychological and physiological problems, risk of errors and decreased quality of work [62].

Although only five per cent of the sample were classified as having a healthy diet, almost three-quarters of participants believed that their diet was healthy. This apparent dissonance between beliefs about their health, and reported behaviour has been shown elsewhere in both registered [7] and pre-registered samples [16].

Participants associated negative changes to their diet with commencement of their training, and this was attributed to academic pressures, adapting to shift work during placement and financial constraints. Other studies with pre-registered nurses and midwives have associated poor dietary habits with a demanding workload and financial constraints [16]. In registered nurse populations, seven similar barriers to healthy eating have consistently emerged from the literature, including: lack of time/overwork, lack of resources/facilities, fatigue, outside commitments, ‘unhealthy’ food culture, supportive versus unsupportive individuals, and positive versus negative role models and are mirrored in this study [63,64,65,66]. Yet, it may not be diet alone on night shifts that has an impact as it has been shown that on average night workers’ energy intake is the same as day workers, although day workers tend to have a higher protein intake. The factors that seem to play a role in the negative role of night shifts on one’s health and diet include increased snacking of unhealthy foods [67], presence of metabolic syndrome [68] and more frequent chronic fatigue [69,70]. There are multiple other factors involved that still need further investigation [57,71].

Education and training around healthy lifestyles are important, although education alone may not be sufficient to improve diet and obesity in nurses and midwives [50,72,73]. Approximately 40% of pre-registered nurses in a previous study agreed that their nursing training had negatively impacted their own health behaviours [16]. Increasing motivation to make lifestyle changes will be important, since motivation for healthy lifestyles has been reported to be low in both pre-registered [73] and registered nurses [74]. A review of interventions to improve health showed that interventions focused solely on education might be less likely to result in positive outcomes than interventions targeting behavioural change [6]. Furthermore, studies in other settings have identified that nurses often perceive more barriers to healthy lifestyle behaviours than facilitators, and institutional (or even societal) changes may be required to shift health behaviours rather than simply intervening at an individual level to improve diet and other health behaviours [65].

Heterogeneity within professional roles makes it difficult to identify a single reason for obesity and also to develop one-size-fits-all approach to health promotion [7]. The UK Royal College of Nursing (RCN) ‘Healthy Workplace, Healthy You’ and Royal College of Midwifery (RCM) ‘Caring for You’ campaigns demonstrate the commitment of professional bodies towards the health and wellbeing of nurses and midwives. The RCN has cited the lack of healthy eating programmes for nurses and stressed the need for increased availability of healthy foods on night shifts, alongside regular breaks [75]. In 2019, the UK government announced a review into hospital food combined with the ‘sugar tax’, which aimed to improve access to healthy food for both patients and staff overnight; however, the implementation of this and the effectiveness of this on staff dietary choices and weight is contentious and is yet to be reviewed [76].

Workplace staff champions are advocated [77], with a role to encourage healthy eating, healthy behaviours and being a source of support to staff for health behaviour change. In the case of students, this role could be incorporated into mentor responsibilities, although this would require exploration into the feasibility of adding this to the workload of busy staff. It may be more prudent to consider peer-to-peer support initiatives, and the implementation of health and wellbeing champions in educational settings. This may provide developmental opportunities for healthcare students, and opportunities for students in these roles to experience engagement in health promotion activities by advocating healthy lifestyle behaviours and positive coping strategies among their peers.

A prior review suggested there is a lack of strong evidence for specific successful interventions that reduce the levels of overweight and obesity in nursing and midwife populations [78]. However, a recent systematic review of lifestyle interventions for nurses found that interventions targeting diet or body composition (as well as physical activity and stress) are more likely to have positive outcomes for nurses’ health and/or wellbeing than other types of lifestyle health promotion interventions [6]. Healthcare workers’ dietary habits (as well as other lifestyle behaviours) may be influenced by organisational or job-related factors; for example, a study in South Korea [79] showed that collegial workplace relationships were associated with more healthy eating among nurses. The authors propose that nursing performance quality and patient outcomes may be improved through enhancing nurses’ personal resources and responsibility for their own health (at an individual level) but only while providing adequate staffing and resources (at the organisational level). This highlights the importance of organisational support for individual-level change.

### 4.2. Role Modelling

Promoting healthy lifestyle behaviours is advocated as part of the professional role of nurses and midwives [8,80]. This is generally accepted by pre-registered nurses [81], where it has previously been shown that 96% of pre-registration nurses (compared with 98.6% of nurses/midwives here) agreed that health promotion was part of their role [31]. Although findings with registered healthcare professionals are somewhat mixed, prior research with nurses showed that 75% believed health promotion to be part of their role [74,82], but many were unaware of the guidance that states that they should be healthy role models and there are few definitions of what this entails. One study by Darch and colleagues [83] suggested that definitions of healthy role models include: being caring, non-judgemental, trustworthy, inspiring and motivating, self-caring, knowledgeable and self-confident, innovative, professional and having a deep sense of self—yet structured undergraduate education on this is lacking. In a later study, both pre-registered and registered nurses maintained that hospitals and placements were not nurturing environments for healthy lifestyles and education did not support them to be healthy role models [15]. Some pre-registered nurses reported that their own beliefs on role modelling behaviours were impacted by that of their registered peers who lacked the time to help them develop health promotional skills [81].

There is much debate over the importance of nurses and midwives ‘practicing what they preach’ [30,84]. The present study found that 84.7% and 76.9% of participants felt that they should maintain a healthy diet and weight respectively, and similar findings have previously been reported in studies with pre-registered nurses [16,73]. The present study also found that overweight or obese participants were less likely to hold this view, a finding echoed in previous research [16]. As discussed previously, many pre-registered nurses comment that they feel hypocritical giving weight loss advice if they are themselves overweight [15]; conversely, some have stated an increased rapport if they were themselves overweight or had experienced weight loss [19,20]. The views of patients play an important role in this debate—since patients may be less likely to heed the advice of healthcare professionals who do not ‘practice what they preach’ and this is explored in detail in Section 4.3.

Engagement in health promotion practice is an important factor. Our study found that those who had personally engaged in health promotion were more likely to think it was part of their professional role. Previous studies have also found that regular engagement with health promotion increased confidence in delivering it, especially when delivered in a structured and supportive clinical teaching setting. This demonstrates the importance of regular involvement in the practice, although clearer guidance to students about the specific public health interventions to be promoted is required [26,82].

Of some concern is that a large proportion of participants in this study who reported having had no engagement in health promotion were in their final year of study. This may be because registered nurses are too busy to support students in developing this practice or it may be due to insufficient opportunity; for example, in clinical settings where there is less involvement in health promotion [81]. For example, nurses on placement in general practice surgeries reported time constraints from short appointment slots, a lack of proven long-term health promotion programs to refer patients to and difficulties in raising topics if a patient is not willing to discuss it [85]. Development was also hindered if mentors were ‘ignorers’, i.e., they would not discuss with patients topics they found difficult to broach, such as obesity [85]. Observing this could potentially lead to negative experiences of placements, and negative attitudes towards health promotion practice for nurses and midwives in training.

Improvement strategies that are focused on effective leadership and management are at the heart of inciting change. This includes nurses and midwives being advocates for protected breaks, promoting healthy behaviours and lifestyle choices, and being beacons for change on behalf of their profession; this is likely to have the follow-on effect of increasing health promotion to patients and nurses/midwives fulfilling their function as role models [79,86,87]. This aligns with the notion that health professionals recognise that their own lifestyle choices impact on care quality [88]. Interventions to address obesity in pre-registered nurses and midwives need to consider the complexity of combining study with clinical placements, and the barriers to engaging in workplace health promotion programmes that are often identified by registered healthcare professionals [78] such as shift work, lack of breaks, the fast-paced nature of the job, and emotional labour.

### 4.3. Attitudes towards Health Promotion (ATHPP)

This study aimed to ascertain whether there is a relationship between body composition and ATHPP. Higher reported BMI was a predictor for reduced ATHPP and participants’ confidence that patients would follow their advice. This is congruent with previous findings. Prior research has shown that pre-registered nurses with an unhealthy lifestyle (and lower self-esteem) held a more negative health promotion attitude [88]. Further studies show that overweight or obese professionals are less likely to give health advice on weight management and that patients may be less likely to trust that message [24,75].

Building on the discussion in Section 4.2, it is widely disputed whether nurses and midwives should be role models. It has been found that nurses and midwives report being less able to promote healthy lifestyle behaviours that they do not follow themselves [25,89], a belief mirrored in pre-registered nurses [16,73], both potentially believing themselves to be hypocritical [26,89]. There is emerging evidence for the influence of nurses’ apparent health status on patient perceptions of nurses and engagement with health advice. For example, patients in one study reported having less confidence in obese nurses to provide health promotion [90]. It has been shown that less than 1 in 10 patients would take advice on diet and lifestyle from an overweight doctor, and employees in other areas of the public sector who are a healthy weight are three times more likely to encourage positive healthy behaviours than those who are overweight or obese [91]. Other reports show that professionals with a healthy BMI are more successful at helping patients achieve clinically significant weight loss (52% vs 29% in high BMI professional), although here there was no difference in perceived trust in advice regardless of practitioner’s BMI [92]. Trust in health professionals is a recurring topic in the literature and has been shown to be important in health promotion activities. For example, patients show increased satisfaction, positive health behaviours and a higher quality of life when there is a level of trust between the patient and the clinical worker across a diverse range of health settings [93]. Trust has also been cited by midwives as a key facilitator for promoting health promotion messages [94]. Therefore it is interpersonal skills that reinforce the trust building qualities of good communion respect and non-judgment [18] that should underpin education in the pre-registration phase of health promotion.

There is an awareness among healthcare professionals that obesity can negatively impact patient care and there may be poor support for empowerment of patients to lose weight [95]. However, the relationship between the patient and the nurse/midwife is viewed to be an important factor when approaching health promotion on weight management [96] with overweight or obese healthcare professionals perhaps being more relatable to patients [16] since they have personal experience of weight management issues [89]. Studies have shown that health professionals with a higher BMI may be more likely to hold a positive attitude towards their obese patients [20], whereas other nurses and midwives may stigmatise them; their personal overweight can help build a rapport with patients [19]. Many health professionals state that they draw from personal experience rather than evidence-based practice when giving health promotion advice and as a result, a health professional who has not undergone weight loss may be perceived to be less suited to giving advice to an overweight or obese patient [97]. Interestingly, dietary habits are often ignored as a contributing factor to health promotion, and, indeed, it was not a significant predictor for ATHPP in this study. In one study, registered nurses typically displayed a highly variable knowledge regarding weight loss practices and only 25% advised reducing calorie intake as a tool [98]. Nurse education about diet and weight loss may need improvement and can be addressed in the pre-registration phase and through continual education.

Our study identified a clear relationship between exposure of pre-registered nurses and midwives to health promotion during their training and their attitudes to health promotion practice and identified differences between these pre-registered groups. This warrants further discussion around the nature of health promotion practice for nurses and midwives. Nurses are well placed to deliver health promotion to others, as they see patients at a time when they are receptive to health promotion advice, when in need of treatment or in pain and this is often more effective than when the advice is given as a preventative measure. Not only does this place nurses in an ideal position, but it also strengthens the case for structured health promotion materials and education to be made available to staff in these locations [99]. While nurses may be ideally placed to deliver health promotion messages, obesity prevention and management requires a multidisciplinary approach, requiring contextual knowledge. It therefore requires an integrated education and framework to be developed that encompasses both staff and the environment in which the message is delivered [100]. Similarly, it is difficult for nurses with the already acknowledged pressures on their workload to identify those in need of intervention. In one study, healthcare professionals did not deliver health promotion information in 50% of interactions where there was a perceived need, citing a lack of interpersonal or training skills on delivering the message as the reason [101]. Screening programs on all patients admitted to wards may allow health information to be highlighted in a routine manner to patients and health promotion frameworks to be delivered in a targeted manner and should be researched further [102].

Pre-registered midwives in the current study reported significantly more experience in health promotion than pre-registered nurses. Conversely, research has reported that hospital midwives feel more averse to giving health promotion as they believe a labour-related hospital stay is typically too short to develop a relationship within which they can deliver effective health promotion [89]. Community placements are deemed to have a greater link to health promotion [81] and community health practitioners tend to have a greater rapport, a significant and lengthy instructive experiences and in depth knowledge on their subjects [103] while being in a position of trust which has been shown to be key in imparting health promotion information [94]. Therefore, the greater prevalence of community placements in midwifery may explain a greater exposure to health promotion amongst this pre-registered group. Yet, many pregnant women are dissatisfied with the weight control advice they receive. Again a lack of confidence, time pressures, poor health promotion education and materials have been cited by midwives, with a lack of equipment availability to support high BMI pregnancies and attempts to treat all patients the same, leading to the normalisation of high BMI pregnancies [94,104]. There is scope to investigate further the level of preparedness for health promotion practice in pre-registered nurses/midwives and to develop further learning opportunities for health promotion observation and practice within pre-registration education programmes and clinical placements.

The present study identified many barriers to delivering dietary health promotion as part of weight management. These include a negative RA and a lack of engagement in health promotion, overweight/obesity, and shift work. Common barriers raised included time pressures, as well as inadequacies in education and training in health promotion practice. The perceived lack of clear guidelines and preparation leads to a reduced confidence in delivering advice on health promotion; health promotion should therefore be featured more prominently in student curriculums and teaching should include workshops for communication skills and mentorship in clinical settings [73,87,105]. Since RA is a predictor for ATHPP, it is vital that nurses and midwives connect to the expectation of professional bodies that health promotion is part of their role, and enhanced education in this area could help to foster a positive RA. This could be introduced at a very early stage, such as orientation/induction to the course or clinical placements by tutors or mentors (e.g., inclusion of structured discussion on effective coping strategies and managing self-care and healthy lifestyles around studies and placements).

Wills and Kelly [7] suggest creating a specific health promotion domain in training; however, this already features in the UK Nursing and Midwifery Council standards for competence [12]. The way in which behaviour change methods are taught may need refining. The MECC [105] health policy is not a new one, however 31.4% of professionals claim to have never heard of it [101]. There is also a need for clarity around which patient groups would benefit most from health promotion advice, since it has been proposed that this would allow for more targeted health promotion behaviours, which may help to reduce the burden on over-worked nurses [101].

The healthcare workplace has a huge impact on pre-registered placements and registered nurses and midwives; education alone is less effective in changing health behaviours and consideration of the organisational influence on health behaviours is paramount [6]. However, organisational solutions are complex and unique to each workplace setting, with further evidence needed before a single strategy can be implemented. Nevertheless, there is a clear economic argument for promoting health in healthcare employees [106]. Prior studies have successfully implemented workplace wellness programmes that are accessible to nurses, midwives and other healthcare workers [107,108], including general health checks [109], physical activity interventions [110,111] and digital educational interventions to promote nurses’ and midwives’ health and wellbeing [112,113]. Again, this highlights the scope for development of peer-to-peer interventions to facilitate healthy lifestyle choices in healthcare professionals; nurses and midwives are both targets and facilitators of health promotion and have previously been advocated as ideal workplace health champions in healthcare organisations [77]. It has been purported that training in workplace health should be mandatory for all frontline staff [112]; this should include reflection on personal lifestyle choices since formal educational programmes tend to be limited to health promotion practice with patients but do not always include a translation of this knowledge to one’s own behaviour and lifestyle choices. Health and wellbeing for nurses and midwives should be an integral part of pre-registration education programmes, as well as continuing professional development.

More structured methods of identifying those in need of health promotion are needed, such as screening and clear guidelines on what to deliver and how to deliver it are still needed system wide. This may help to identify opportunities for pre-registered nurses and midwives to observe or practice health promotion, and for mentors to have the capacity to support this. It is suggested that nurses’ and midwives’ working environments have a greater influence on their lifestyle choices than their knowledge of healthy behaviours [114]. A wealth of evidence supports this view: shift work can negatively affect BMI [7,63,64,115]. This is not compensated for by the nature of the job role which comprises mostly light-intensity physical activity [116] and healthy food is not easily accessible to hospital staff on night shifts [115]. While shift work is an unavoidable part of the job role, organisational changes, such as increasing accessibility to healthy food, could ensure a healthier workforce [73,78] and, in turn, a more positive ATHPP.

#### Strengths and Limitations

This study adds to a limited evidence-base around the perceptions of nurses and midwives towards their own weight status and diet, and their attitudes towards health promotion and their professional role. These findings are timely in the context of the current coronavirus SARS-2 (COVID-19) pandemic, during which the pressure on health and care organisations has been immense, and there is increasing attention to the physical and mental health and wellbeing of the workforce. A healthy workforce is of paramount importance for the future sustainability of health and care services. Self-care (alongside structural, organisational and job-related factors) is one aspect of health and well-being, that is strongly advocated in supportive interventions that have arisen during the COVID-19 pandemic. For example, a digital support package developed for health and care workers and trainees [117,118] emphasises the value of aspects of self-care such as diet, exercise, shift-work and sleep, all of which are known to be related to overweight and obesity.

It is predicted that overestimation of height and underestimation of weight can lead to inaccurate classification of BMI groups in 8.4–16.6% of cases [54], although this indicates that overweight and obesity levels could actually be underestimated in our sample. One of the strengths in this study was the use of free-text comments about the factors affecting one’s diet, which provided unique and interesting information. Nevertheless, findings are based on self-reported data from pre-registered healthcare professionals from a single institution, which limits the generalisability of our findings. Future studies would benefit from incorporating participants from other settings and contexts. Comparisons across cultural groups, geographical regions and countries would be of value. While the response rate was low, this is commonly observed in online surveys [119] and our response rate was higher than that obtained in other online surveys with healthcare student samples [120]. Our sample included pre-registered healthcare professionals from both nursing and midwifery across all years of study and was broadly representative of the nursing and midwifery population at the participating institution with regards demographic profile, albeit a smaller sample of midwives than nurses. This is, however, just a first step towards a better understanding of dietary habits and their role in health promotion across broader settings. Lastly, we need to stress that due to the brevity of our survey, we were not able to include data on social desirability, or other health or social indicators that could have added to the richness of our findings.

## 5. Conclusions

Overweight and obesity are prevalent in pre-registered nurses and midwives, and very few consume a healthy diet, which they commonly attribute to their mood status, the cost of healthy foods and working shifts on placements. Those with a healthy weight and a healthy diet, and those who have more positive attitudes towards role modelling, were more likely to have positive attitudes towards health promotion. Having more experience of practicing health promotion, consuming a healthy diet and being a healthy weight predicted greater confidence that patients would heed their health promotion advice. However, previous health promotion engagement negatively affected one’s attitude to health promotion practice, suggesting that health promotion experiences may not be meeting students’ expectations; this warrants further exploration.

Universities need to focus on increasing health promotion and access to services and facilities for nursing and midwifery students (as well as tutors, and mentors), to help them adopt/maintain healthy behaviours. Alongside wider health promotion initiatives, this might include advocating rest breaks and self-care, increasing access to fruit and vegetables onsite, providing weight management advice or support, incorporating brief 10-min exercise sessions into the working day, offering strength training and/or education on back injury prevention, and so on. Nursing and midwifery education programmes should include training on workplace health and wellbeing and the value of translating health promotion education into self-care, alongside guidance and structural/organisational supports for the adoption of healthy behaviours around the schedules of university and placements. Importantly, there must be consideration of the individual, interpersonal and environmental factors that influence health choices, and management of the structural and job-related factors that can be obstacles to, or enable, health behaviours. Peer health champions could help with advocacy and support, thus gaining health promotion and health signposting experiences. Mentors could facilitate observation and more active involvement of students in health promotion during placements to foster positive attitudes to health promotion practice. This should include opportunities to check students’ expectations and assist them in reflecting on health promotion experiences.

## Figures and Tables

**Table 1 ijerph-18-13419-t001:** Reported and perceived weight status in body mass index classification.

Reported Weight *n* (%)		Perceived Weight *n* (%)
BMI (kg/m^2^)	*n* = 346	*n* = 346
Underweight (≤18.4)	15 (4.3)	12 (3.5)
Healthy Weight (18.5–24.9)	214 (61.8)	225 (65.0)
Overweight (25.0–29.9)	85 (24.6)	94 (27.2)
Obese (≥30)	32 (9.2)	15 (4.3)

The two outcomes were significantly correlated (Spearman’s r = 0.80, *p* < 0.001).

**Table 2 ijerph-18-13419-t002:** Factors influencing dietary habits.

	Total Sample *n* (%)	Unhealthy Diet (≤2.5) *n* (%)	Healthy Diet (>2.5) *n* (%)	*p*
Stress	271 (78.3)	189 (79.7)	82 (75.2)	0.34
Mood	269 (77.7)	189 (79.7)	80 (73.4)	0.19
Boredom	207 (59.8)	137 (57.8)	70 (64.2)	0.26
Money	197 (56.9)	149 (62.9)	48 (44.0)	0.001
Drinking Alcohol	127 (36.7)	86 (36.3)	41 (37.6)	0.81
Worry	170 (49.1)	126 (53.2)	44 (40.4)	0.027
Exam Pressure	135 (39.0)	102 (43.0)	33 (30.3)	0.024
Watching Television	50 (14.5)	39 (16.5)	11 (10.1)	0.12
Lack of Time	15 (4.3)	12 (5.1)	3 (2.8)	0.33
Health	6 (1.7)	2 (0.8)	4 (3.7)	0.08 ^#^
Lack of Knowledge	2 (0.6)	2 (0.8)	0 (0)	1.0 ^#^
Peer Pressure	1 (0.3)	0	1 (0.9)	0.32 ^#^

Chi-square test used to compare proportions between groups, with Bonferroni correction (unless ^#^ used); ^#^ Fisher’s exact test used.

**Table 3 ijerph-18-13419-t003:** Linear regression model predicting RA (*n* = 343).

Variable	B	SE	β	*p* Value	95% CI
Constant	2.62	0.29	-	<0.001	2.05–3.18
Age	−0.002	0.004	−0.03	0.59	−0.01–0.01
Gender	0.06	0.10	0.03	0.53	−0.14–0.26
Ethnicity	0.10	0.08	0.07	0.22	−0.06–0.26
Year of study	0.03	0.03	0.06	0.31	−0.03–0.09
BMI	−0.01	0.01	−0.06	0.30	−0.02–0.01
HP engagement	0.23	0.05	0.33	<0.001	0.14–0.33
HP observation	−0.07	0.04	−0.10	0.13	−0.15–0.02
Healthy diet	0.06	0.06	0.05	0.32	−0.06–0.18

RA= role attitudes; BMI Body Mass Index; HP Health promotion; gender was dummy coded as 0 = female, 1 = male; ethnicity was dummy coded as 1 = white British/Irish/other, 2 = other (non-White).

**Table 4 ijerph-18-13419-t004:** Linear regression model predicting ATHPP (*n* = 343).

Variable	B	SE	β	*p* Value	95% CI
Constant	1.32	0.26	-	<0.001	0.81–1.83
Age	−0.005	0.003	−0.05	0.20	−0.01–0.002
Gender	0.19	0.08	0.09	0.02	0.02–0.35
Ethnicity	0.003	0.07	0.002	0.97	−0.13–0.13
Year of study	0.01	0.02	0.02	0.60	−0.04–0.06
BMI	−0.01	0.01	−0.08	0.078	−0.02–0.001
HP engagement	−0.18	0.04	−0.25	<0.001	−0.26–−0.10
HP observation	0.04	0.04	0.06	0.24	−0.03–0.11
Healthy diet	−0.02	0.05	−0.02	0.64	−0.12–0.08
RA	0.69	0.04	0.67	<0.001	0.60–0.78

ATHPP= attitudes towards health promotion practice; BMI Body Mass Index; HP Health promotion; gender was dummy coded as 0 = female, 1 = male; ethnicity was dummy coded as 1 = white British/Irish/other, 2 = other (non-White).

**Table 5 ijerph-18-13419-t005:** Linear regression model predicting confidence in patients valuing HP advice (*n* = 343).

Variable	B	SE	β	*p* Value	95% CI
Constant	1.32	0.40	-	<0.001	0.52–2.11
Age	0.002	0.005	0.02	0.67	−0.01–0.01
Gender	0.20	0.13	0.08	0.12	−0.05–0.45
Ethnicity	0.14	0.10	0.07	0.17	−0.06–0.35
Year of study	−0.05	0.04	−0.07	0.20	−0.12–0.03
BMI	−0.02	0.01	−0.16	0.003	−0.04–−0.01
HP engagement	0.19	0.06	0.20	0.002	0.07–0.31
HP observation	0.09	0.06	0.10	0.12	−0.02–0.20
Healthy diet	0.17	0.08	0.11	0.03	0.02–0.32
RA	0.19	0.07	0.14	0.006	0.06–0.33

HP = health promotion; BMI Body Mass Index; RA Role attitudes; gender was dummy coded as 0 = female, 1 = male; ethnicity was dummy coded as 1 = white British/Irish/other, 2 = other (non-White).

**Table 6 ijerph-18-13419-t006:** Linear regression model predicting HP behaviour (*n* = 343).

Variable	B	SE	Beta	*p* Value	95% CI
Constant	1.25	0.27	-	<0.001	0.71–1.79
Age	−0.001	0.004	−0.02	0.76	−0.01–0.01
Gender	−0.10	0.09	−0.06	0.25	−0.27–0.07
Ethnicity	0.05	0.07	0.04	0.75	−0.09–0.19
Year of study	−0.02	0.04	−0.04	0.45	−0.07–0.03
BMI	0.01	0.01	0.05	0.30	−0.01–0.02
HP engagement	0.26	0.04	0.39	<0.001	0.18–0.34
HP observation	0.01	0.04	0.01	0.85	−0.07–0.08
Healthy diet	0.03	0.05	0.03	0.59	−0.08–0.13
RA	0.16	0.05	0.17	0.001	0.07–0.25

HP = health promotion; BMI Body Mass Index; RA Role attitudes; gender was dummy coded as 0 = female, 1 = male; ethnicity was dummy coded as 1 = white British/Irish/other, 2 = other (non-White).

## Data Availability

The data presented in this study are available on request from the corresponding author.

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
