# Peer review of "Obesity and Diet Predict Attitudes towards Health Promotion in Pre-Registered Nurses and Midwives"

_ijerph, 2021, doi:10.3390/ijerph182413419_

Round 1

Reviewer 1 Report

I congratulate the authors on their work. It was a relevant and very well conducted investigation. The text presented is very well written, with a satisfactory theoretical basis. The statistical analysis is well elaborated and described and the references are current and have been very well-located.

My only consideration to make is for the content below.

  • line 156/157 - "Finally, participants were asked to list the key factors that influenced their dietary habits. Free text response boxes were included to enable participants to expand on their answers". Considering the wide variety of possible answers in this item, how have the factors been defined? And how were they determined in statistical analysis?

Author Response:

Response: Thank you for taking the time to review, this is appreciated. 

Comment: I congratulate the authors on their work. It was a relevant and very well conducted investigation. The text presented is very well written, with a satisfactory theoretical basis. The statistical analysis is well elaborated and described and the references are current and have been very well-located.

Response: We want to thank the reviewer for such a positive and kind comment.

Comment: My only consideration to make is for the content below.

line 156/157 - "Finally, participants were asked to list the key factors that influenced their dietary habits. Free text response boxes were included to enable participants to expand on their answers". Considering the wide variety of possible answers in this item, how have the factors been defined? And how were they determined in statistical analysis?

Response: We have elaborated on this point more.

“Free-text comments were analysed using thematic analysis (Braun and Clarke, 2006), employing an inductive approach in which coding and theme development were driven by the content of the comments. This involved analysis of semantic content of the entire free-text data (ie, whether comments contained references to behaviours, psychological, social or environmental factors) and whether comments were of a positive or negative nature. A coding taxonomy was produced for sorting qualitative data into categories of participant experience, developed by one researcher (KW) in collaboration with a further researcher (HB). Once the main taxonomy had been established (ie, it accommodated the majority of comments without need for additional categories), the coding of comments was checked by two researchers (KW and HB) and any conflicts resolved by discussion. Individual categories were then subjected to a second stage of more detailed sorting into subcategories. For example, behavioural factors were categorised into ‘drinking alcohol’ and ‘watching tv’; psychological factors were categorised into ‘stress’, ‘mood’, ‘worry’, and so on. Categories/subcategories were subjected to cross-sectional analysis to highlight the frequencies of common themes and allowed for comparison across groups that reported healthy versus unhealthy diet (Table 2)”.

Reviewer 2 Report

This article is a good contribution to the literature on obesity and diet among students preparing to be nurses and midwives. Given that the obesity rate was moderate for the age group, the tone of the abstract/introduction could be more neutral. Now it seems blaming/stigmatizing towards students. The authors could better justify the study by being concerned that as early adults, this rate of obesity bodes poorly for their health over their lifetime, perhaps compromising their ability to be effective on the job. The authors make the point that being obese compromises their ability to be effective health educators and they can more softly make the point of role modeling in that section. The authors should also note that with long shifts and high expectations/requirements that are placed on nurses and midwives that increase stress levels, cortisol and other metabolic factors may also predispose them to obesity. They mention but could be clearer that positive coping techniques could be part of student orientation. Similarly irregular or lack of good sleeping patterns can also attribute to obesity.
In the discussion section, the authors could be more imaginative in solutions directed towards the workplace and schools. For example schools could incorporate brief (10-15 minute) exercises as part of every class to add up to 40 minutes each day. Some strength training would help prevent back injuries that plague nurses when they move patients or lift heavy objects. Having schools and workplaces build these in would benefit all workers AND be a good role model for other businesses and settings. Similarly schools and workplaces could make fruit and vegetables available as free snacks in their settings. Again this would allow the school/workplace to be a good role model for other clinical or practice sites and would benefit faculty/staff (another group that is likely to be obese and perhaps bad role models for students/mentees). For the strengths/limits section. One strength is the use of the free text which provided unique and interesting information well used in analysis. They should note that the brevity of the survey did not allow for other variables to be tested like social desireability, other health indicators or social variables. Finally, the sample is not "broadly representative" as claimed by the authors. At best it is a first step towards what should be further research across settings to verify the findings. Although the discussion is lengthy, it is a good review of existing literature.

Author Response

Response: Thank you for taking the time to review, this is appreciated. 

Comment: This article is a good contribution to the literature on obesity and diet among students preparing to be nurses and midwives. Given that the obesity rate was moderate for the age group, the tone of the abstract/introduction could be more neutral. Now it seems blaming/stigmatizing towards students.

Response:

  • We apologise for this and did not intend for the language to be stigmatizing. We have reviewed the text again and believe that we have been factual and state prevalence rates alongside expectations of regulating bodies, and what we view to be a balanced presentation of the controversial debate around nurses/midwives as roles for health (including the pro’s and con’s) of alternative views.
  • We have reviewed the abstract which reports our findings. There is little that can be altered in the abstract without misrepresenting our rationale and results, but we have made some wording changes. We have changed the word ‘expected’ to ‘advised’ in the rationale in attempt to soften the language, although we believe the original wording ‘expected’ was accurate since nurses and midwives are indeed expected to be role models by their governing bodies, and one of our arguments is that awareness of this expectation is low. In our recommendations for practice, we have changed ‘should’ to ‘could’, and ‘facilitate opportunities’ to ‘facilitate more opportunities’.
  • In exploring the rates of overweight / obesity in healthcare students compared with university students more generally, we have actually found that healthcare students (ie, nursing, midwifery) have similar or even higher rates of overweight and obesity compared with university students generally – in England and 22 other countries. Therefore, we have added these details to the introduction to strengthen our rationale. However, to recognise that overweight/obesity is common in university years we have reflected on this and listed some of the factors that might influence this:

“Weight gain during university years is common [32] and has been linked to poor eat-ing habits, stress and lower physical activity following the transition from a structured secondary education environment to university [33,34]. However, despite higher levels of education and training related to public health and health promotion, healthcare students appear to have similar or even higher rates of overweight and obesity ([16]: 24%; [29]:44%; [30] 27.1%; [31]:28%) compared with university students more broadly. For example, in [35], 22% of students were overweight or obese, (22 countries, n=15,746); in [32], 14.5% of female students were overweight or obese (from 23 univer-sities in England). This is important since weight gain in younger years has been linked to overweight and obesity in adulthood [36]], which poses a significant threat to individual health and wellbeing [1–3], and the costs associated with obesity are linked to workplace injury, lost productivity and sickness absence [37,38] which may impact on health and care services once pre-registered individuals join the health workforce.”

We have also demonstrated that the challenges align with those of registered nurses, and further highlighted the importance of job-related factors to avoid focusing only on individual behaviours and choices:

“Interventions to address obesity in pre-registered nurses and midwives need to consider the complexity of combining study with clinical placements, and the barriers to engaging in workplace health promotion programmes that are often identified by registered healthcare professionals [Kelly and Wills, 2018], such as shift work, lack of breaks, the fast-paced nature of the job, and emotional labour.

Comment: The authors could better justify the study by being concerned that as early adults, this rate of obesity bodes poorly for their health over their lifetime, perhaps compromising their ability to be effective on the job.

Response: The link between obesity and health is outlined at the outset and threaded through the discussion. We have added in an additional statement in the introduction:

This is important since weight gain in younger years has been linked to overweight and obesity in adulthood [Guo et al, 2002], which poses a significant threat to health and wellbeing [1-3], and the costs associated with obesity are linked to workplace injury, lost productivity and sickness absence [Ferrie et al, 2007; van Duijvenbode et al, 2009]

Comment: The authors make the point that being obese compromises their ability to be effective health educators and they can more softly make the point of role modeling in that section.

Response: The section overall includes much discussion about role modeling. However, we have amended the text to make it more explicit within this particular paragraph:

“However, overweight or obese health professionals also report lower confidence levels and feelings of hypocrisy while delivering weight management advice [15], which may compromise their ability to promote health effectively and suggests that role modeling may be important for patient care. The authors should also note that with long shifts and high expectations/requirements that are placed on nurses and midwives that increase stress levels, cortisol and other metabolic factors may also predispose them to obesity.”

Comment: The authors should also note that with long shifts and high expectations/requirements that are placed on nurses and midwives that increase stress levels, cortisol and other metabolic factors may also predispose them to obesity.

Response: We included this point in the discussion:

“Shift work and the desynchronisation of circadian rhythms may have adverse effects on nurses’ health - a systematic review (Rosa et al, 2019) showed that night work involves greater risk of cardiovascular and metabolic diseases compared to daytime work, which may predispose them to obesity. Indeed, shift work has numerous risks for nursing staff including psychological and physiological problems, risk of errors and decreased quality of work (Rosa et al, 2019)”.

Comment: They mention but could be clearer that positive coping techniques could be part of student orientation.

Response: We have now referred to the inclusion of role modeling in curricula and orientation within the introduction, with examples of strategies:

“This suggests that role modeling may be important for patient care and strategies for achieving this (e.g., time management, self-care, healthy eating, exercise, managing shifts and sleep) could be more prominent in healthcare curricula, and orientation to courses and placements by academic tutors and mentors.”

We have also referred to positive coping strategies in the discussion:

“This could be introduced at a very early stage, such as orientation / induction to the course or clinical placements by tutors or mentors (e.g., inclusion of structured discussion on effective coping strategies and managing self-care and healthy lifestyles around studies and placements).”

Comment: Similarly irregular or lack of good sleeping patterns can also attribute to obesity. 

Response: We have ensured that this is covered in the discussion:

“Short sleep duration along with other dimensions of poor sleep has been associated with obesity both cross-sectionally and longitudinally [Ogilvie and Patel, 2017]. Shift work and the desynchronisation of circadian rhythms may have adverse effects on nurses’ health - a systematic review (Rosa et al, 2019) showed that night work involves greater risk of cardiovascular and metabolic diseases compared to daytime work, which may predispose them to obesity. Indeed, shift work has numerous risks for nursing staff including psychological and physiological problems, risk of errors and decreased quality of work (Rosa et al, 2019).”

Comment: In the discussion section, the authors could be more imaginative in solutions directed towards the workplace and schools. For example, schools could incorporate brief (10-15 minute) exercises as part of every class to add up to 40 minutes each day. Some strength training would help prevent back injuries that plague nurses when they move patients or lift heavy objects. Having schools and workplaces build these in would benefit all workers AND be a good role model for other businesses and settings. Similarly schools and workplaces could make fruit and vegetables available as free snacks in their settings. Again this would allow the school/workplace to be a good role model for other clinical or practice sites and would benefit faculty/staff (another group that is likely to be obese and perhaps bad role models for students/mentees).

Response: Thank you for these excellent ideas, we have included them in the conclusion:

“Universities need to focus on increasing health promotion and access to services and facilities for nursing and midwifery students (as well as tutors, and mentors), to help them adopt/maintain healthy behaviours. Alongside wider health promotion initiatives, this might include advocating rest breaks and self-care, increasing access to fruit and vegetables onsite, providing weight management advice or support, incorporating brief 10-minute exercise sessions into the working day, offering strength training and/or education on back injury prevention, and so on. Nursing and midwifery education programmes should include training on workplace health and wellbeing and the value of translating health promotion education into self-care, alongside guidance and structural supports for the adoption of healthy behaviours around the schedules of university and placements. Importantly, there must be consideration of the individual, interpersonal and environmental factors that influence health choices, and management of the structural and job-related factors that can be obstacles to, or enable, health behaviours.”

Comment: For the strengths/limits section. One strength is the use of the free text which provided unique and interesting information well used in analysis.

Response: Thank you for this positive comment. We have added it in the text.

One of the strengths in this study was the use of free-text comments about the factors affecting one’s diet, which provided unique and interesting information.

Comment: They should note that the brevity of the survey did not allow for other variables to be tested like social desirability, other health indicators or social variables.

Response: We have added this point to the limitation section.

“Lastly, we need to stress that due to the brevity of our survey, we were not able to include data on social desirability, or other health or social indicators that could have added to the richness of our findings.”

Comment: Finally, the sample is not "broadly representative" as claimed by the authors. At best it is a first step towards what should be further research across settings to verify the findings.

Response: We meant that the sample was broadly representative of the nursing and midwifery student population of the institution where the study took place (according to our demographics for respondents), but we understand that this could have been clearer. We have reworded that point.

“Our sample included pre-registered healthcare professionals from both nursing and midwifery across all years of study and was broadly representative of the nursing and midwifery population at the participating institution with regards demographic profile, albeit a smaller sample of midwives than nurses. This is however just a first step towards better understanding of dietary habits and their role for health promotion across broader settings. Incorporating samples from other contexts would be of high value”.

Comment: Although the discussion is lengthy, it is a good review of existing literature.

Response: We appreciate this positive feedback.

Reviewer 3 Report

ABSTRACT

This study explores whether obesity and eating habits are related to attitudes towards healthy role modeling and the practice of health promotion.

DISCUSSION
There is little incorporation of discussion with other studies related to the subject of this study. For the most part, the authors limit themselves to commenting on their results, but do not contrast them with those of other studies. It is necessary to incorporate more scientific evidence in this section.
BIBLIOGRAPHIC REFERENCES
It would be necessary not to include references older than 10 years.

Author Response

Response: Thank you for taking the time to review, this is appreciated.

Comment:

ABSTRACT

This study explores whether obesity and eating habits are related to attitudes towards healthy role modeling and the practice of health promotion.

DISCUSSION
There is little incorporation of discussion with other studies related to the subject of this study. For the most part, the authors limit themselves to commenting on their results, but do not contrast them with those of other studies. It is necessary to incorporate more scientific evidence in this section.

Response:

  • In line with the comments made by reviewer 2 (‘a good review of existing literature’) we respectfully do not agree with this point. We have already provided a very lengthy discussion of our results, and its connection with other relevant literature. However, in addressing some specific comments made by other reviewers, additional references have been added throughout the revision process that may help to allay any concerns of reviewer 3.
  • We have also set the findings in the context of supportive intervention (which includes self-care, diet, obesity) that has been developed in response to the current covid-19 pandemic, to ensure the findings of this work are set in the current context and are noted as timely:

“These findings are timely in the context of the current coronavirus SARS-2 (COVID-19) pandemic, during which the pressure on health and care organisations has been im-mense, and there is increasing attention to the physical and mental health and wellbe-ing of the workforce. A healthy workforce is of paramount importance for the future sustainability of health and care services. Self-care (alongside structural, organisational and job-related factors) is one aspect of health and well-being, that is strongly advocat-ed in supportive interventions that have arisen during the COVID-19 pandemic. For example, a digital support package developed for health and care workers and trainees [118,119] emphasises the value of aspects of self-care such as diet, exercise, shift-work and sleep, all of which are known to be related to overweight and obesity.”

Comment:  BIBLIOGRAPHIC REFERENCES
It would be necessary not to include references older than 10 years.

Response: Respectfully, we do not agree with this comment, this is not a rule of thumb to limit oneself to last 10 years of literature. The vast majority of the literature is more recent, yet adding an arbitrary date cut-off for literature would not allow us to reference key studies that are still relevant today and support our rationale or discussion. We believe that this point is also demonstrated by reviewer 1 who commented on our work: “the references are current and have been very well-located.”

Reviewer 4 Report

Novel research 

Methodology and statistics clearly described.  

While I think it would have been interesting to see the trends between pre-registered nurses and midwives, given the different in sample size I agree with your decision to leave the data as one set.

I appreciate your recognition of the limitations of self-reported data.  Also data from a single institution limits the ability to generalize over a broader population.

Author Response

Response: Thank you for taking the time to review, this is appreciated.

Comment: Novel research 

Response: Thank you for noticing that.

Comment: Methodology and statistics clearly described.  

Response: Thank you for this positive comment.

Comment: While I think it would have been interesting to see the trends between pre-registered nurses and midwives, given the different in sample size I agree with your decision to leave the data as one set.

Response: Thank you for understanding the limitations of our sample and agreeing with our decision.

Comment: I appreciate your recognition of the limitations of self-reported data.  Also, data from a single institution limits the ability to generalize over a broader population.

Response: Thank you for your comment. We have now expanded on the point about the single institution more in the discussion (limitations section).

“Nevertheless, findings were based on self-reported data from pre-registered healthcare professionals from a single institution, which limits the generalisability of our findings. Future studies would benefit from incorporating more institutions, and comparison across cultural groups, geographical regions and countries would be of value

Reviewer 5 Report

Dear authors,

I think that this work is of interest and is well executed.

Nevertheless, I have some comments.

p. 9 lines 349-352: Please cite some of the factors that may be involved in the impact of night shifts on health outcomes and diet.

p. 10 Section i. Role Modelling: While reading this section, it feels that the part about the concept of "practice what you preach" from the point of view of the patient is missing. Please provide some key points here and point out that it will be analysed in depth in the next section.

Author Response

Response: Thank you for taking the time to review, this is appreciated.

Comment: Dear authors, I think that this work is of interest and is well executed.

Response: Thank you for this positive feedback.

Comment: Nevertheless, I have some comments.

p.9 lines 349-352: Please cite some of the factors that may be involved in the impact of night shifts on health outcomes and diet.

Response: Thank you for this suggestion. We have now added examples of factors that may be involved in the impact of night shifts on health outcomes and diet, and provided multiple new references to support this:

“The factors that seem to play a role in the negative role of night shifts on one’s health and diet include increased snacking of unhealthy foods (Ulusoy et al, 2021), presence of metabolic syndrome (Samhat et al, 2020), more frequent chronic fatigue (Ferry et al, 2016; Pietroiusti et al, 2010).”

Comment: p.10 Section i. Role Modelling: While reading this section, it feels that the part about the concept of "practice what you preach" from the point of view of the patient is missing. Please provide some key points here and point out that it will be analysed in depth in the next section.

Response: Thank you for this and we agree, it would be helpful to acknowledge patients in this section so that the reader does not assume we have not considered this. We have included an additional statement in section i and referred to section ii for more detail. We have kept this statement succint to avoid repetition of information between sections i and ii, since the discussion is already quite long:

“The views of patients play an important role in this debate – since patients may be less likely to heed the advice of healthcare professionals who do not ‘practice what they preach’ and this is explored in detail in section ii.”